# CONTROLLING ZERO-SHOT AGENTS VIA INVERSE INVERSE PLANNING

## ABSTRACT

Zero-Shot Reinforcement Learning (ZSRL) trains agents to solve tasks that are not explicitly encountered during training. While recent ZSRL methods demonstrate impressive generalization capabilities, the interpretability of their zero-shot behaviors remains largely unaddressed. This poses a challenge for real-world deployment in safety-critical domains such as autonomous driving and assistive robotics. In this paper, we propose a novel integration of *Inverse Inverse Planning* (IIP)—a behavior modification technique inspired by narrative analogies in storytelling—into the ZSRL setting. Our approach enables users to remove specific task-level intentions from a zero-shot policy without additional retraining. The result is a modified agent whose behavior is easier to inspect, explain, and control. We demonstrate that IIP can selectively suppress undesired behaviors in new tasks while preserving performance on the original task, offering a new direction for interpretable and controllable generalization in ZSRL.

## 1 INTRODUCTION

Zero-Shot Reinforcement Learning (ZSRL) has emerged as a promising paradigm for training agents that generalize to novel tasks without explicit fine-tuning (Touati et al., 2022; Jeen et al., 2024; Sun et al., 2025). Such generalization is critical in real-world applications (e.g., autonomous vehicles), where it is infeasible to anticipate every possible scenario during training (Levine et al., 2020; Dulac-Arnold et al., 2021; Kiran et al., 2021). Despite recent advances, Deep Reinforcement Learning (DRL) continues to suffer from a key limitation: its lack of interpretability. As noted by Glanois et al. (2022) and Zahavy et al. (2016), policies learned by DRL are generally difficult to understand due to the black-box nature of deep neural network architectures.

According to Glanois et al. (2022), existing work on interpretable RL falls into three main categories. The first focuses on *interpretable transition models*, which aim to learn human-readable representations of environment dynamics, using probabilistic approaches such as decision trees and graphical models (Degris et al., 2006; Kansky et al., 2017), physics-based or graph-based deterministic models (Scholz et al., 2014; Zhang et al., 2018), or structured neural networks (Li et al., 2015; Battaglia et al., 2016; Finn et al., 2016). The second line of research seeks to develop *interpretable preference models*, making reward functions more transparent by expressing them in the form of decision trees (Srinivasan & Doshi-Velez, 2020), logical rules (Aksaray et al., 2016; Littman et al., 2017; Li et al., 2017), finite-state machines or "reward machines" (Toro Icarte et al., 2019; Xu et al., 2020; Gaon & Brafman, 2020), or Boolean algebra task compositions (Nangue Tasse et al., 2020). The third direction, *interpretable decision-making*, attempts to make an agent's policy itself more interpretable, for example by learning directly interpretable policies (Ernst et al., 2005; Likmeta et al., 2020), approximating black-box policies with simpler surrogates (Liu et al., 2018; Verma et al., 2018), or incorporating interpretability through architectural design (Tang et al., 2020; Mott et al., 2019; Annasamy & Sycara, 2019).

While these directions improve transparency at the model or task-specification level, none address the challenge of making agents *visually interpretable* to humans. In particular, an agent's raw behavior (e.g., performing a jump) should be readily understandable to observers without requiring additional post-hoc explanation. A natural solution to this problem was proposed in the seminal work "Acting as Inverse Inverse Planning" (Chandra et al., 2023), which posits that agents should behave in ways that maximize the probability that observers infer a specific intended goal. This

approach, termed **Inverse Inverse Planning (IIP)**, builds upon the framework of **Inverse Planning (IP)** (Baker et al., 2009; Ullman et al., 2009; Tauber & Steyvers, 2011; Zhi-Xuan et al., 2020). In IP, an observer infers an agent's latent goal from its observed behavior by computing the posterior: $P(g \mid \text{actions}) \propto P(\text{actions} \mid g) P(g)$, where $g$ denotes a candidate goal. IP has been widely used in cognitive science, human–robot interaction, and multi-agent systems to explain and predict behavior through goal inference.

In contrast, IIP inverts the perspective: instead of observers inferring goals from behavior, agents deliberately choose behaviors that maximize the probability that observers infer a desired goal. This makes IIP a powerful tool for crafting interpretable and communicative behaviors (Chandra et al., 2023).

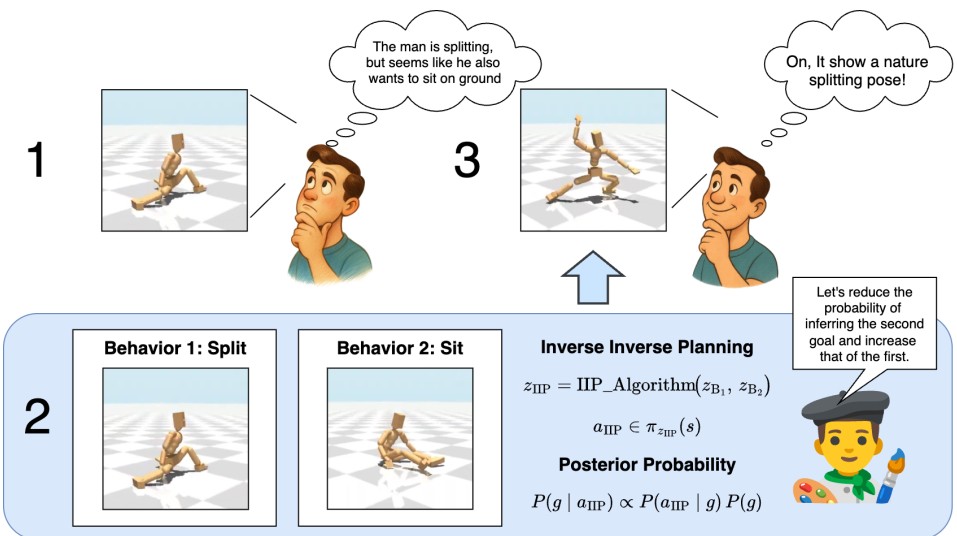

Figure 1: Illustration of how IIP improves behavior realism and interpretability. Although ZSRL agents can achieve high rewards (zone 1), their behaviors may appear ambiguous or physically implausible to human observers. By removing latent components associated with undesirable behaviors (e.g., zone 2), IIP refines the task embedding to yield more natural and expressive motions (zone 3). The mathematical formulation of $\pi_{z_{\text{IIP}}}(s)$ is given in Eq. equation 2. Here, $z$ denotes a latent task embedding, as commonly used in ZSRL (Jeen et al., 2024). *Note:* IIP$_{\text{algorithm}}$ refers to our proposed method (see Appendix A.2), and $B_1$ and $B_2$ refer to the two behaviors, respectively.

To illustrate this motivation, Figure 1 contrasts the behavior of a zero-shot agent before and after applying IIP. While the original behavior achieves high rewards, it often appears ambiguous or physically implausible. By modifying the task vector to suppress latent components tied to undesired behaviors, IIP yields behavior that is more natural, expressive, and aligned with human expectations—enhancing interpretability and trust.

However, applying IIP in ZSRL is nontrivial. IIP requires optimizing agent behavior with respect to the inverse planning model, which involves repeatedly computing posterior distributions over goals and adjusting actions accordingly. This creates significant computational challenges, especially in high-dimensional goal spaces. A naive implementation of IIP is thus computationally infeasible for real-world RL agents.

To address this, we propose a **lightweight reformulation of IIP** tailored to ZSRL. In ZSRL, tasks are defined via *latent embeddings* (Barreto et al., 2017; Touati & Ollivier, 2021; Touati et al., 2022; Jeen et al., 2024; Sun et al., 2025), allowing agents to generalize without retraining. We reinterpret goal inference in terms of these latent representations, avoiding the need to evaluate full posterior distributions. Based on this reformulation, we introduce a **bisection-based search procedure** that efficiently modifies the latent task vector to suppress undesired secondary goals. Concretely, we refine the primary task vector by subtracting a scaled projection of the secondary task vector, with the scaling factor tuned via bisection. This approach avoids combinatorial overhead while preserving IIP's interpretability and control benefits.

Our key contributions are threefold. First, we introduce a post hoc IIP refinement procedure for pre-trained ZSRL agents, which suppresses behaviors associated with undesired secondary tasks while preserving competence on the primary task. Second, we evaluate IIP on the Metamotivo Humanoid benchmark (Tirinzoni et al.), testing over 100 task pairs across 5 random seeds each. IIP reduces unintended behavior in 76% of cases while maintaining or improving primary task performance. Third, we showcase novel applications of IIP, including removing boundary-violating behaviors, enhancing narrative coherence in storytelling agents, and improving physical plausibility in humanoid animation. Overall, our experiments validate the core objective of IIP: modifying the task vector leads to more interpretable and goal-aligned agent behavior—where the probability of inferring the intended goal increases, and that of the unintended goal decreases.

## 2 RELATED WORK

### 2.1 ZERO-SHOT REINFORCEMENT LEARNING

Traditional Reinforcement Learning (RL) algorithms are primarily designed for online settings, where agents continuously interact with the environment to gather experience and update their policies. However, in many real-world applications—such as healthcare, autonomous driving, and robotics—such interaction can be prohibitively expensive, risky, or altogether infeasible (Levine et al., 2020; Fu et al., 2021). These limitations have motivated the development of ZSRL paradigms, where the agent must learn from static datasets and generalize to unseen tasks or environments without any additional online exploration (Touati et al., 2022; Jeen et al., 2024).

To address the challenge of generalization in RL, several lines of research have emerged. One such direction is *goal-conditioned RL*, where policies are trained to reach a specified goal state from any initial state (Eysenbach et al., 2022; Ma et al., 2022; Yang et al., 2023; Wang et al., 2023; Park et al., 2023). Another is *model-based RL*, which aims to learn task-independent world models that can generalize across tasks by capturing the underlying environment dynamics (Chua et al., 2018; Hafner et al., 2020). *Multi-task RL* extends this by explicitly training agents on multiple related tasks to encourage the emergence of generalizable behavior (He et al., 2023; Lan et al., 2023). Finally, *unsupervised skill discovery (USD)* focuses on learning diverse and discriminative behaviors in reward-free settings, thereby facilitating downstream adaptation to new tasks (Eysenbach et al., 2018; Laskin et al., 2022). None of the above methods can achieve zero-shot generalization, as explained in Sun et al. (2025).

Typical ZSRL algorithms rely on a number of architectural building blocks and training paradigms that aim to support such generalization without online finetuning. These include *successor representations* (Dayan (1993), Grimm et al. (2019)), *universal function approximators* (Schaul et al., 2015), *successor features* (Barreto et al., 2017; Zhang et al., 2017; Borsa et al., 2018; Hansen et al., 2019), and *successor measures* (Blier et al., 2021). These are commonly instantiated via either the *Universal Successor Features (USF)* framework (Barreto et al., 2017) or *Forward-Backward* (FB) architectures (Touati & Ollivier, 2021).

Successor features factorizes the value function into two components: a dynamics-dependent successor representation and a reward-dependent weight vector. As the original paper puts it, "a good representation for a state would be one that resembles the representations of its successors." This decoupling of environment dynamics from rewards allows the agent to generalize across tasks that share the same dynamics but differ in reward functions. USF scales this concept to high-dimensional environments by leveraging deep neural networks. It introduces a state embedding function $\phi(s)$ and conditions predictions on a family of policies $\pi_z$, approximated by $\pi_z(s) = \arg\max_a \psi(s, a, z)^\top z$, where $\psi(s, a, z)$ represents the successor feature. USFs are typically trained using temporal difference (TD) learning.

The FB framework uses two encoders: a forward dynamics encoder $F(s, a, z)$ mapping state-action-task tuples to a latent space, and a backward encoder $B(s')$ projecting the next state into the same space. Training aligns $F(s, a, z) \approx B(s')$, while action selection maximizes $F(s, a, z)^\top z$. A limitation of FB is its tendency to choose out-of-distribution (OOD) actions, leading to unsafe behavior. To mitigate this, Jeen et al. (2024) introduced *Value-Conservative FB (VCFB)* and *Measure-Conservative FB (MCFB)*, which penalize OOD actions via value regularization and visitation measures, respectively. Sun et al. (2025) extended FB to online unsupervised RL through *Dual-Value*

*FB (DVFB)*, which combines skill value $Q_M = F(s, a, z)^\top z$ and exploration value $Q_P$, leveraging RND and contrastive learning. A reward mapper later aligns external rewards with learned skills, improving zero-shot adaptation.

## 2.2 Inverse and Inverse-Inverse Planning

Inverse Planning (IP) refers to the problem of inferring an agent's latent goal given its observed actions. This concept was first introduced by Baker et al. (2009), who proposed casting action understanding as IP. Since then, IP has been applied to diverse domains, including analyzing social interactions (Tauber & Steyvers, 2011; Ullman et al., 2009) and enabling machines to recognize when humans fail to achieve their goals (Zhi-Xuan et al., 2020).

Once we can model how humans infer goals from behavior, the natural next step is to design agents that choose actions to influence such inferences—what we call *Inverse Inverse Planning (IIP)*. The goal of IIP is for an agent to act in a way that causes an observer to infer a particular intended goal. This flips the IP setup: rather than observing actions to infer goals, the agent selects actions to steer the observer's inferences.

The concept of IIP has roots in the graphics and vision communities. Durand et al. (2002) introduced the notion of *inverse-inverse rendering*, where visual depiction (e.g., a painting) is seen as a process of influencing an observer's perception by optimizing over their inverse-rendering model. Subsequently, Kukkonen (2014) proposed re-examining narratives from a Bayesian, probabilistic perspective. This abstraction was further developed by Chandra et al. (2022), who proposed using inverse-inverse rendering to generate visual illusions by modeling and manipulating human visual inference. Building on this foundation, Chandra et al. (2023) introduced IIP in the context of story-telling agents. Their framework enables agents to take actions not just for achieving task outcomes, but to better align with an audience's expectations and mental models. This leads to behavior that is more interpretable and compelling from the observer's perspective. A related application in RL is explored by Strouse et al. (2019), who proposed a method for learning to either share or hide intentions using information regularization. Their framework supports strategic behavior generation in cooperative and competitive multi-agent environments, without requiring explicit access to the world model or direct modeling of other agents. These works collectively motivate the emerging field of IIP, which seeks to unify communication, intent modeling, and action selection into a cohesive framework grounded in human-centered inference.

## 3 Preliminary

### 3.1 Reward-Free Markov Decision Process

In unsupervised reinforcement learning, a *reward-free* Markov Decision Process (MDP) is defined as $\mathcal{M} := (\mathcal{S}, \mathcal{A}, P, \rho_0, \gamma)$, where $\mathcal{S}$ and $\mathcal{A}$ denote the state and action spaces, $P : \mathcal{S} \times \mathcal{A} \to \mathcal{D}(\mathcal{S})$[1] is the transition probability function mapping state-action pairs to distributions over next states, $\rho_0 \in \mathcal{D}(\mathcal{S})$ is the initial state distribution, and $\gamma \in (0, 1)$ is the discount factor. The set of Markovian policies is defined as $\Pi := \{\pi \mid \pi : \mathcal{S} \to \mathcal{D}(\mathcal{A})\}$, where $\pi(a \mid s)$ denotes the probability of taking action $a$ in state $s$. Given a reward function $R : \mathcal{S} \times \mathcal{A} \to \mathbb{R}$, the state-action value function under policy $\pi$ is defined as $Q^\pi_{\mathcal{M}}(s, a) := \mathbb{E}\left[\sum_{t=0}^{\infty} \gamma^t R(s_t, a_t) \mid s_0 = s, a_0 = a, \pi\right]$, where the expectation is taken over trajectories generated by starting from the initial state-action pair ($s_0 = s, a_0 = a$), and following the policy $\pi$ thereafter. That is, $a_t \sim \pi(\cdot \mid s_t)$, and $s_{t+1} \sim P(\cdot \mid s_t, a_t)$. We use $\Pr(\cdot \mid s_0, a_0, \pi)$ and $\mathbb{E}[\cdot \mid s_0, a_0, \pi]$ to denote the probability and expectation over the trajectory $(s_t, a_t)_{t \geq 0}$ induced by executing policy $\pi$ starting from the initial pair $(s_0, a_0)$.

### 3.2 Forward–Backward Representation

The FB representation builds on the concept of the *successor measure*. The successor measure captures the discounted cumulative occupancy of states $s_{t+1}$, conditioned on the initial pair $(s_0, a_0)$

---

[1]$\mathcal{D}(\mathcal{S})$ denotes the space of probability distributions over $\mathcal{S}$.

and policy $\pi$. Formally, it is defined as:

$$\mathcal{M}^\pi(s_0, a_0, X) := \sum_{t=0}^\infty \gamma^t \Pr(s_{t+1} \in X \mid s_0, a_0, \pi), \quad \forall X \subseteq \mathcal{S}. \tag{1}$$

The FB representation approximates this measure using two learned functions: $F : \mathcal{S} \times \mathcal{A} \times Z \to \mathbb{R}^d$, $\quad B : \mathcal{S} \to \mathbb{R}^d$, defined over a latent task space $Z \subseteq \mathbb{R}^d$. The approximation satisfies:

$$\begin{cases} M^{\pi^z}(s_0, a_0, X) \approx \int_X F(s_0, a_0, z)^\top B(s)\, \rho(ds), & \forall s_0 \in \mathcal{S}, a_0 \in \mathcal{A}, X \subseteq \mathcal{S}, z \in \mathbb{R}^d, \\ \pi_z(s) \approx \arg\max_{a \in \mathcal{A}} F(s, a, z)^\top z, & \forall s \in \mathcal{S}, z \in \mathbb{R}^d. \end{cases} \tag{2}$$

When Equation 2 holds, the optimal action-value function for any reward can be directly computed as: $Q_r^*(s, a) = F(s, a, z_r)^\top z_r$, and the corresponding optimal policy is given by: $\pi_{z_r}(s) = \arg\max_{a \in \mathcal{A}} F(s, a, z_r)^\top z_r.$. To obtain the embedding $z_r$ for any bounded reward function $r : \mathcal{S} \times \mathcal{A} \to \mathbb{R}$, we define:

$$z_r = \mathbb{E}_{(s,a) \sim \rho}\left[r(s,a)B(s)\right], \tag{3}$$

where $\rho$ is a fixed distribution over state-action pairs (e.g., the dataset distribution in offline RL). In summary, the FB representation encodes a *latent representation of tasks*, allowing reward functions to be embedded into vectors $z_r$ that define behavior. This enables efficient, planning-free policy execution by composing the learned representations $F$, $B$, and the reward embedding $z_r$.

### 3.3 INVERSE INVERSE PLANNING

IIP is a concept that arises from studies of storytelling and social cognition. Humans often interpret others' actions through *Inverse Planning (IP)*—that is, by inferring an agent's goals based on observed behavior using Bayesian reasoning: $P(g \mid \text{actions}) \propto P(\text{actions} \mid g)P(g)$, where $g$ is a goal hypothesis (e.g., "the robot is helpful") entertained by an observer. IIP reverses this reasoning process: instead of directly optimizing for rewards or goals, the agent selects behaviors that increase the likelihood that an observer will infer a *desired interpretation* of its intent or values. For instance, rather than taking the shortest path, an agent might deliberately choose a longer but more expressive trajectory to reveal its intended goal or demonstrate cooperative intent as in Chandra et al. (2023). This formulation captures the idea of *performative behavior*—where actions are chosen not only for functional effectiveness but also for their interpretability to humans. IIP is especially relevant in human-robot interaction, explainable AI, and multi-agent collaboration settings where modeling the observer's inferences is essential.

## 4 METHOD: COMBINING INVERSE INVERSE PLANNING WITH FORWARD–BACKWARD PLANNING

Our method combines Forward–Backward (FB) embeddings with Inverse Inverse Planning (IIP) to enable post hoc behavior shaping of pretrained zero-shot agents. Figure 2 provides an overview of the IIP–FB pipeline. The process begins with a **backward encoder** that infers task vectors $z_1$ and $z_2$ from offline trajectories for two tasks: the *desired* task and an *undesired* secondary task. These are used to construct a candidate IIP vector: $z_{\text{iip}} = z_1 - \lambda z_2$, where $\lambda$ controls the extent to which undesired behaviors are suppressed. To find the optimal $\lambda$, we apply a bisection search that evaluates rollouts from the policy $\pi_{z_{\text{iip}}}$ using two reward functions $r_1$ and $r_2$. The search continues until the expected reward for the undesired task falls below a threshold, while preserving competence on the desired task. This pipeline enables agents to maintain internal reward grounding (via $z_1$) while modifying outward behavior to align with audience-inferred goals, thereby improving interpretability and controllability. See Appendix A.2 for full pseudocode.

## 5 EXPERIMENTS

We conduct four experiments to evaluate different aspects of our IIP method. First, we quantitatively demonstrate that optimizing the latent task vector $z$ in a secondary environment (Environment 2)

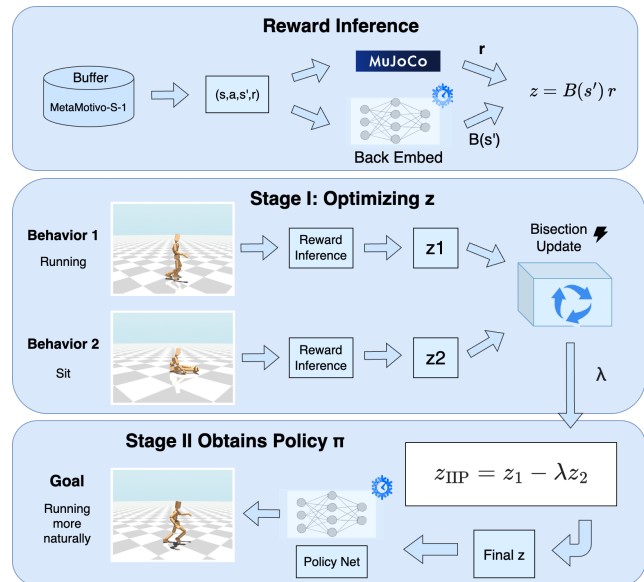

Figure 2: Overview of the IIP–FB pipeline. We extract task vectors using FB reward inference, optimize $z_{\text{iip}}$ via bisection (Stage I), and execute policies for goal inference (Stage II). *(Snowflake indicates frozen modules; thunder indicates the bisection update is fast and gradient-free.)*

reliably reduces the corresponding reward, indicating that IIP can successfully discover a new task embedding $z_{\text{iip}}$ that inverts unintended behaviors. The remaining experiments are qualitative: we (i) remove unwanted behaviors to enforce stricter boundaries while maintaining task completion; (ii) refine behavior to improve interpretability and storytelling—e.g., preventing sitting during walking to suggest aversion to bugs and evoke emotional resonance; and (iii) enhance physical plausibility by adjusting animations to appear biomechanically natural. Together, these results highlight IIP's ability to support reward suppression, behavior editing, narrative control, and motion realism.

### 5.1 QUANTITATIVE REWARD SUPPRESSION

We begin by quantitatively evaluating the ability of our method to suppress behaviors associated with an undesired task. Specifically, we demonstrate that the final task vector $z_{\text{iip}}^{\text{final}}$, produced by our IIP pipeline, yields a reduced reward in Environment 2 compared to the initial vector $z_1$, which is inferred solely from the primary task. This suppression serves as a proxy for removing unwanted behaviors from the agent's policy. All experiments in this section and those that follow utilize the pretrained humanoid agents provided by Metamotivo (Tirinzoni et al.), chosen for their high-fidelity and naturalistic movement, which make them well-suited for nuanced behavioral analysis. Further details on the Metamotivo environment are provided in Appendix A.3 (environment description), Appendix A.5 (body-part specifications), and Appendix A.6 (reward formulations).

We constrain the Lagrange multiplier $\lambda$ to lie within $[\lambda_{\min}, \lambda_{\max}] = [-1, 1]$ to prevent excessive deviations from the original task embedding, thereby preserving the overall behavior while allowing sufficient flexibility for suppression. The threshold for behavior suppression in Environment 2 is set to 50% of the initial reward, striking a balance between aggressive behavior removal and preservation of agent competence. We set the tolerance parameter to 0.1 to allow early stopping once sufficient suppression is achieved, thus improving computational efficiency. Finally, the maximum number of optimization steps is capped at 100, which empirically provides sufficient iterations for convergence in most task pairs without unnecessary overhead.

For each of the 46 predefined tasks in the Metamotivo benchmark, we randomly sample 3 other tasks to act as secondary (undesired) tasks using a fixed seed of 42. Each resulting task pair is evaluated over 5 random seeds (0–4), and we report: (i) the reward of the original task vector $z_1$ in Environment 1 (i.e., the intended task setting); (ii) the reward of $z_1$ in Environment 2 (i.e., the undesired task setting); and (iii) the reward of the modified vector $z_{\text{iip}}^{\text{final}}$ in both environments. These

metrics jointly assess the degree of reward suppression and task retention achieved through our IIP optimization. As shown in Figure 3, we find that IIP preserves the original task's reward in Environment 1 in the majority of cases (**65.2%** of task pairs saw no drop), while suppressing reward in Environment 2 in a significant fraction of relevant cases (**76.5%** of task pairs with non-zero baseline reward in Environment 2 showed a reduction after IIP).

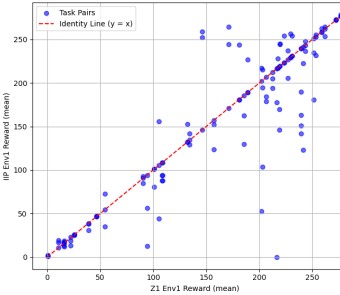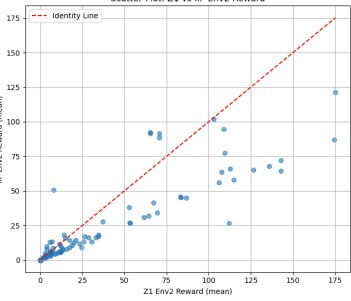

Figure 3: Scatter plots comparing mean rewards of $z_1$ (x-axis) versus $z_{\text{iip}}^{\text{final}}$ (y-axis). **Left:** Environment 1 (original task). **Right:** Environment 2 (undesired task). The dashed red line indicates $y = x$ (no change). Points above the line represent reward preservation or improvement, while points below the line indicate degradation. IIP generally preserves performance in Environment 1 while suppressing reward in Environment 2.

In Table 1 and Table 2, we highlight representative task pairs based on the signed percent change in Environment 2 reward: the best cases are those where IIP sharply reduces reward in Environment 2, whereas failure cases correspond to reward increases in Environment 2. In contrast, Table 3 and Table 4 sort task pairs by the absolute percent change to surface the most impactful differences regardless of direction—showcasing the most suppressed rewards (best absolute cases) as well as the most amplified ones (failure absolute cases). The percentage $\Delta$ is computed using the difference in mean rewards between IIP and baseline $z_1$: $\Delta = 100 \times \frac{\text{IIP\_mean} - z_1\_mean}{|z_1\_mean|}$. Standard deviation across 5 random seeds is shown in parentheses.

Table 1: Top-5 task pairs with strongest Environment 2 suppression (by % change). Rewards are averaged over 5 seeds.

| Task1 | Task2 | $z_1$ Env 1 | IIP Env 1 | $z_1$ Env 2 | IIP Env 2 | %$\Delta$ Env 2 |
|---|---|---|---|---|---|---|
| sitonground | move-ego-low-0-0 | 201.78(13.35) | 52.48(44.14) | 112.13(7.98) | 26.75(19.17) | -76.14 |
| move-ego-low-90-0 | move-ego-low-0-0 | 239.31(2.49) | 150.83(37.45) | 24.66(2.96) | 9.22(4.61) | -62.61 |
| move-ego-low-0-2 | move-ego-low-0-0 | 239.49(14.67) | 162.83(35.96) | 30.70(4.88) | 13.43(3.89) | -56.25 |
| lieonground-down | crawl-0.4-0-d | 203.05(2.94) | 103.78(30.73) | 142.72(0.22) | 64.29(19.94) | -54.95 |
| headstand | move-ego-low-0-0 | 54.71(14.68) | 72.75(11.69) | 12.05(3.91) | 5.72(1.75) | -52.53 |

Table 2: Top-5 task pairs where IIP failed (Environment 2 reward increased by %). Rewards are averaged over 5 seeds.

| Task1 | Task2 | $z_1$ Env 1 | IIP Env 1 | $z_1$ Env 2 | IIP Env 2 | %$\Delta$ Env 2 |
|---|---|---|---|---|---|---|
| raisearms-l-l | move-ego-90-2 | 272.33(0.19) | 273.08(0.11) | 7.93(0.47) | 50.73(5.06) | +539.72 |
| headstand | move-ego-90-4 | 54.71(14.68) | 35.24(12.86) | 3.92(0.85) | 9.75(0.66) | +148.72 |
| rotate-x–5-0.8 | move-ego-90-2 | 22.24(4.07) | 13.10(3.20) | 5.77(0.65) | 12.76(0.89) | +121.14 |
| lieonground-up | move-ego-90-2 | 216.19(1.07) | 177.86(7.23) | 3.82(0.91) | 7.86(0.60) | +105.76 |
| split-0.5 | move-ego-90-2 | 251.20(3.19) | 234.50(7.18) | 6.90(1.57) | 13.44(0.82) | +94.78 |

The results demonstrate that IIP can effectively suppress secondary behaviors (as measured by reward in Environment 2) in many task pairs. For top-performing pairs, we observe over 70% reduction in Environment 2 reward, confirming IIP's potential for targeted behavioral inhibition. Nonetheless, several *failure cases* exist where IIP fails to suppress the undesired behavior and instead *amplifies* it—sometimes by more than 500% in relative terms. These outliers suggest that some task embeddings are highly entangled, making disentanglement via linear constraint-based methods difficult. In particular, task pairs that share overlapping movement primitives or constraints

Table 3: Top-5 task pairs with strongest Environment 2 suppression (by absolute reward drop).

| Task1 | Task2 | $z_1$ Env 1 | IIP Env 1 | $z_1$ Env 2 | IIP Env 2 | $\Delta$ Env 2 |
|---|---|---|---|---|---|---|
| move-ego-low-90-2 | move-ego-90-2 | 239.31(2.49) | 141.87(5.55) | 174.57(2.93) | 87.19(1.70) | -87.38 |
| sitonground | move-ego-low-0-0 | 201.78(13.35) | 52.48(44.14) | 112.13(7.98) | 26.75(19.17) | -85.38 |
| lieonground-down | crawl-0.4-0-d | 203.05(2.94) | 103.78(30.73) | 142.72(0.22) | 64.29(19.94) | -78.43 |
| crawl-0.4-0-d | move-ego-low-0-0 | 218.88(12.18) | 169.85(18.72) | 142.74(8.28) | 72.01(6.09) | -70.73 |
| move-ego-low-180-2 | move-ego-90-2 | 105.25(5.20) | 155.85(11.92) | 136.02(2.26) | 67.80(1.09) | -68.22 |

Table 4: Top-5 task pairs where IIP amplified Environment 2 reward (by absolute increase).

| Task1 | Task2 | $z_1$ Env 1 | IIP Env 1 | $z_1$ Env 2 | IIP Env 2 | $\Delta$ Env 2 |
|---|---|---|---|---|---|---|
| raisearms-l-l | move-ego-90-2 | 272.33(0.19) | 273.08(0.11) | 7.93(0.47) | 50.73(5.06) | +42.80 |
| crawl-0.5-0-u | move-ego-low-0-0 | 108.26(5.00) | 87.82(6.00) | 65.30(0.81) | 92.37(1.81) | +27.07 |
| crawl-0.4-0-u | move-ego-low-0-0 | 108.70(4.52) | 87.81(4.90) | 65.32(0.64) | 91.75(1.63) | +26.43 |
| crawl-0.5-2-u | move-ego-low-0-0 | 16.02(5.50) | 18.49(3.02) | 70.64(3.77) | 91.72(6.83) | +21.08 |
| crawl-0.4-2-u | move-ego-low-0-0 | 15.66(5.75) | 17.77(2.20) | 70.55(3.62) | 88.67(3.51) | +18.12 |

(e.g., posture, joint angles, or locomotion direction) are more susceptible to these failures. Overall, our findings highlight both the strengths and limitations of IIP. While it is generally robust, reliable suppression in challenging cases may require a deeper understanding of task semantics and embedding alignment, motivating future work on more expressive behavior shaping methods. Overall, the quantitative results verify the central claim of IIP: task-vector modification consistently improves interpretability and goal alignment, with increases in intended-goal inference and reductions in unintended-goal attribution. Full results are included in Appendix A.4.

## 5.2 REMOVING UNWANTED BEHAVIORS FROM DEMONSTRATIONS

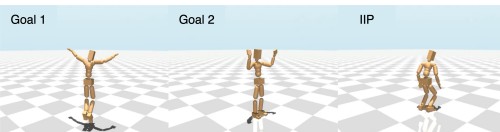

(a) IIP removes unnecessary hand-raising from original jump behavior.
https://tinyurl.com/2dxpj6xk

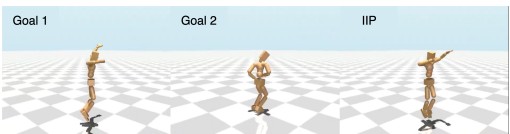

(b) IIP removes unnecessary rotation from original jump behavior, leading to a clean upward motion.
https://tinyurl.com/59umu5kz

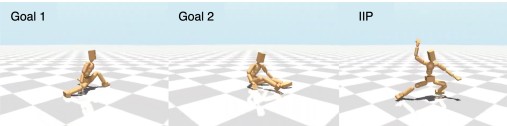

(c) IIP prevents the agent from sitting, emphasizing avoidance of ground contact.
https://tinyurl.com/bdfhju9u

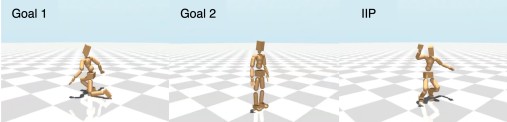

(d) IIP lifts posture constraints, yielding expressive, urgent arm motion.
https://tinyurl.com/u5re55wm

Figure 4: Inverse Inverse Planning (IIP) modifies latent goal embeddings to improve zero-shot generalization across diverse tasks.

As illustrated in Figure 4a, pretrained agents may learn to include stylistic or unnecessary behaviors while completing a task. For example, in some environments, the agent may raise its arms when jumping, even though this motion is irrelevant to the intended task objective. These extraneous behaviors can reduce clarity, increase ambiguity for observers, or lead to misinterpretation of the agent's goal. Another example is shown in Figure 4b, where the agent performs an unnecessary rotational motion during a jump. Through IIP, we can optimize the latent reward embedding to discourage such rotation and generate a revised behavior that better aligns with the intended goal.

### 5.3 Shaping Agent Behavior for Narrative and Interpretability

In some cases, we wish to modify an agent's behavior not merely for functionality, but to convey a narrative or improve interpretability. Consider Figure 4c, where the original agent walks with exaggerated knee bending and occasionally sits on the ground. If the desired narrative is to portray the agent as being afraid of bugs on the ground, then sitting would contradict this interpretation. Using IIP, we remove the undesired sitting behavior, resulting in a revised task embedding $z_{\text{iip}}$ that produces a cautious walk—suggesting the agent is trying hard not to touch the ground. Another example is shown in Figure 4d, where the original agent walks with a low center of gravity and keeps its hands down. To depict the agent as hastily escaping from danger, we remove the behavior that suppresses upward arm movement. The resulting behavior obtained via IIP conveys a more frantic and expressive motion, better aligning with the intended narrative.

### 5.4 Refining Physical Plausibility of Agent Motion

In Figure 5, under the constraints of human biomechanics and physical principles, we observed several instances of unnatural motion. When guided by RL reward functions, the algorithm often prioritizes maximizing the reward signal, which can easily disregard the natural dynamics of real-world human motion. For instance, in Task$_1$, the shin strikes the ground backward at an angle of approximately 30 degrees to propel the body forward, while the slight swinging of the arms appears unnatural and inconsistent with actual human motor patterns. Our method reversely applies the behavior of "sit on ground" to relax the fixation of the limbs. This adjustment produces a gait that conveys the impression of larger, more natural strides during locomotion. In summary, IIP helps restore more interpretable and human-like motion patterns by reducing unnatural behaviors.

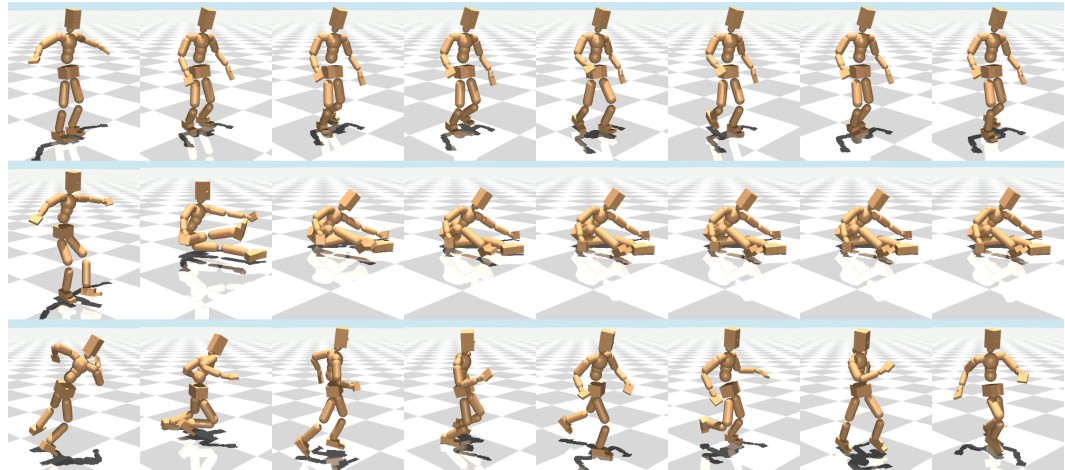

Figure 5: Original agent behavior (top) includes a motion of the humanoid robot during locomotion. By applying IIP and removing the "sit on the ground" motion (middle), it can be observed that the arms and legs exhibit more natural and fluid dynamics, more closely approximating authentic human running behavior (bottom). https://tinyurl.com/47pvmeau

## 6 Conclusion

In this paper, we introduced a novel framework that efficiently integrates Inverse Inverse Planning (IIP) with Zero-Shot Reinforcement Learning (ZSRL). Through extensive evaluations across diverse task pairs, we demonstrated the effectiveness of our approach in resolving behavioral ambiguity and enhancing policy generalization. Beyond performance gains, we also uncovered new applications of IIP, including the ability to improve narrative coherence and physical plausibility in agent behaviors. Future work may explore extending this framework to environments beyond humanoid control, as well as integrating IIP with alternative policy learning architectures.

## REPRODUCIBILITY STATEMENT

All source code necessary to reproduce our results is available at `https://tinyurl.com/bddmx8h5`. Detailed experimental setup is provided in Section 5.1, including hyperparameter choices and evaluation protocols. Appendix A.3 outlines the environment configurations used in our study, while Appendix A.6 describes the reward formulation and computation procedures.

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

# A APPENDIX

## A.1 LLM USAGE

We utilized Large Language Models (LLMs), specifically ChatGPT, solely for polishing the writing in our paper—such as improving grammar and clarity—and for adding documentation and comments to our codebase. No part of the experimental design, algorithm development, or result analysis was generated or influenced by ChatGPT.

## A.2 ALGORITHMIC DETAILS

---
**Algorithm 1** Inverse Inverse Planning (IIP) via Bisection

---
1: **Input:** Reward models `rew_model`$_1$, `rew_model`$_2$; Environments $T_1$, $T_2$; Target reward $R_T$; Tolerance $\epsilon$; Max steps $N$; Bounds $\lambda_{\min}$, $\lambda_{\max}$
2: $z_1 \leftarrow$ `rew_model`$_1$.`reward_inference`$(T_1)$
3: $z_2 \leftarrow$ `rew_model`$_2$.`reward_inference`$(T_2)$
4: **for** $i = 1$ to $N$ **do**
5:     $\lambda \leftarrow (\lambda_{\min} + \lambda_{\max})/2$
6:     $z_{\text{IIP}} \leftarrow z_1 - \lambda \cdot z_2$
7:     $R \leftarrow$ `Rollout`$(T_2, z_{\text{IIP}})$       ▷ Collect rewards by rolling out in $T_2$
8:     **if** $|R - R_T| < \epsilon$ **then**
9:         **break**
10:    **else if** $R > R_T$ **then**
11:        $\lambda_{\min} \leftarrow \lambda$       ▷ Not aggressive enough, shift lower bound up
12:    **else**
13:        $\lambda_{\max} \leftarrow \lambda$       ▷ Too aggressive, shift upper bound down
14:    **end if**
15: **end for**
16: **Output:** Final $z_{\text{IIP}}$

---

## A.3 ENVIRONMENT DESCRIPTION

In this section, we provide a brief overview of the Metamotivo Humanoid Environment (Tirinzoni et al.). The environment is built upon the MuJoCo model of the SMPL agent (Luo et al., 2024; 2023; 2021), with modifications that make the joint ranges and motor controllers more biologically plausible. These improvements are inspired by the CMU humanoid specification used in the `dm_control` suite (Tunyasuvunakool et al., 2020).

The environment supports a wide range of motor control tasks that emphasize different physical capabilities such as locomotion, balance, manipulation, and posture control. Table 5 summarizes the main task categories and their naming conventions. The table is adapted from the original paper (Tirinzoni et al.).

## A.4 FULL RESULTS

In Table 6, we present the complete evaluation results of our experiments using pretrained Metamotivo agents. Each row presents results for a task pair, with IIP applied to influence behavior on the secondary task. Reductions in Environment 2 reward—signifying effective suppression—are highlighted in **bold**.

All values are averaged over 5 random seeds. Numbers in parentheses indicate standard deviation. The percentage change ($\Delta$) is computed over the means:

$$\Delta = 100 \times \frac{\text{IIP\_mean} - z_1\_mean}{|z_1\_mean|}$$

Table 5: Task formats and their corresponding behaviors in the Metamotivo Humanoid Environment.

| Category | Format | Description |
|---|---|---|
| Locomotion | move-ego-[low-]-[angle]-[speed] | Move at a specified heading [angle] and speed [speed]. The "low-" prefix indicates low posture (pelvis constraint); absence implies high posture (head constraint). |
| Standing | move-ego-[low-]-0-0 | Static standing pose. Same posture constraints as locomotion, but with zero velocity. |
| Headstand | headstand | Inverted balance: head down, feet up, with minimal velocity and control effort. |
| Arm Raising | raisearms-[left_pos]-[right_pos] | Raise arms to specified heights: low, med, or high based on wrist z-coordinates. |
| Rotation | rotate-[axis]-[speed]-[height] | Rotate around the x/y/z axis at the desired angular speed, while maintaining height and alignment. |
| Jump | jump-[height] | Reach a target vertical height with sufficient upward velocity. |
| Ground Poses | sitonground, lieonground-[dir], split-[dist] | Sit, lie on ground ([dir]: up/down), or perform splits with a target foot distance [dist]. |
| Crawl | crawl-[height]-[speed]-[facing] | Crawl at specified height and speed; [facing]: down (toward floor) or up (toward sky). |

Table 6: Full evaluation results for all task pairs. Rewards are shown as mean (std) over 5 seeds. Percent change ($\Delta$) is relative to $z_1$.

| Task1 | Task2 | $z_1$ Env 1 | IIP Env 1 | %$\Delta$ Env 1 | $z_1$ Env 2 | IIP Env 2 | %$\Delta$ Env 2 |
|---|---|---|---|---|---|---|---|
| move-ego-0-0 | crawl-0.4-0-d | 276.68 (0.34) | 276.68 (0.34) | 0.00 | 0.00 (0.00) | 0.00 (0.00) | NaN |
| move-ego-0-0 | move-ego-90-4 | 276.68 (0.34) | 278.21 (0.43) | 0.55 | 11.38 (0.60) | 11.69 (5.29) | 2.72 |
| move-ego-0-0 | headstand | 276.68 (0.34) | 276.68 (0.34) | 0.00 | 0.00 (0.00) | 0.00 (0.00) | NaN |
| move-ego-low-0-0 | crawl-0.4-0-d | 189.23 (18.01) | 189.23 (18.01) | 0.00 | 0.00 (0.00) | 0.00 (0.00) | NaN |
| move-ego-low-0-0 | move-ego-90-4 | 189.23 (18.01) | 226.66 (30.77) | 19.78 | 14.32 (1.81) | 18.07 (9.43) | 26.19 |
| move-ego-low-0-0 | headstand | 189.23 (18.01) | 189.23 (18.01) | 0.00 | 0.00 (0.00) | 0.00 (0.00) | NaN |
| headstand | crawl-0.4-0-d | 54.71 (14.68) | 54.18 (7.13) | -0.97 | 1.05 (0.79) | 0.55 (0.42) | **-47.62** |
| headstand | move-ego-90-4 | 54.71 (14.68) | 35.24 (12.86) | -35.59 | 3.92 (0.85) | 9.75 (0.66) | 148.72 |
| headstand | move-ego-low-0-0 | 54.71 (14.68) | 72.75 (11.69) | 32.97 | 12.05 (3.91) | 5.72 (1.75) | **-52.53** |
| move-ego-0-2 | crawl-0.4-0-d | 261.85 (0.29) | 261.85 (0.29) | 0.00 | 0.00 (0.00) | 0.00 (0.00) | NaN |
| move-ego-0-2 | move-ego-90-4 | 261.85 (0.29) | 253.57 (0.80) | -3.16 | 106.08 (0.51) | 56.08 (1.39) | **-47.13** |
| move-ego-0-2 | move-ego-low-0-0 | 261.85 (0.29) | 264.61 (0.39) | 1.05 | 2.89 (1.36) | 1.58 (0.58) | **-45.33** |
| move-ego-0-4 | crawl-0.4-0-d | 253.10 (0.80) | 253.10 (0.80) | 0.00 | 0.00 (0.00) | 0.00 (0.00) | NaN |
| move-ego-0-4 | move-ego-90-4 | 253.10 (0.80) | 231.92 (0.52) | -8.37 | 114.80 (1.53) | 57.85 (0.55) | **-49.61** |
| move-ego-0-4 | move-ego-low-0-0 | 253.10 (0.80) | 255.04 (0.41) | 0.77 | 4.96 (0.82) | 2.50 (0.40) | **-49.60** |
| move-ego–90-2 | crawl-0.4-0-d | 217.31 (3.45) | 217.31 (3.45) | 0.00 | 0.00 (0.00) | 0.00 (0.00) | NaN |
| move-ego–90-2 | move-ego-90-4 | 217.31 (3.45) | 227.93 (2.56) | 4.89 | 1.08 (0.05) | 1.64 (0.20) | 51.85 |
| move-ego–90-2 | move-ego-low-0-0 | 217.31 (3.45) | 216.92 (4.71) | -0.18 | 2.24 (2.44) | 1.53 (1.90) | **-31.70** |
| move-ego–90-4 | crawl-0.4-0-d | 212.26 (2.75) | 212.26 (2.75) | 0.00 | 0.00 (0.00) | 0.00 (0.00) | NaN |
| move-ego–90-4 | move-ego-90-4 | 212.26 (2.75) | 205.15 (4.34) | -3.35 | 34.52 (3.35) | 17.29 (1.61) | **-49.91** |
| move-ego–90-4 | move-ego-low-0-0 | 212.26 (2.75) | 194.16 (3.86) | -8.53 | 3.07 (1.20) | 1.55 (0.60) | **-49.51** |
| move-ego-90-2 | crawl-0.4-0-d | 226.99 (2.15) | 226.99 (2.15) | 0.00 | 0.00 (0.00) | 0.00 (0.00) | NaN |
| move-ego-90-2 | move-ego-90-4 | 226.99 (2.15) | 205.53 (8.27) | -9.45 | 175.13 (0.95) | 121.48 (4.39) | **-30.63** |
| move-ego-90-2 | move-ego-low-0-0 | 226.99 (2.15) | 237.40 (7.25) | 4.59 | 12.94 (15.19) | 8.82 (11.63) | **-31.84** |
| move-ego-90-4 | crawl-0.4-0-d | 185.34 (2.95) | 185.34 (2.95) | 0.00 | 0.00 (0.00) | 0.00 (0.00) | NaN |
| move-ego-90-4 | move-ego-90-2 | 185.34 (2.95) | 129.68 (1.09) | -30.03 | 83.65 (2.56) | 45.21 (1.15) | **-45.95** |
| move-ego-90-4 | move-ego-low-0-0 | 185.34 (2.95) | 162.78 (1.36) | -12.17 | 3.55 (0.37) | 3.58 (0.66) | 0.85 |
| move-ego-180-2 | crawl-0.4-0-d | 219.56 (2.26) | 219.56 (2.26) | 0.00 | 0.00 (0.00) | 0.00 (0.00) | NaN |
| move-ego-180-2 | move-ego-90-2 | 219.56 (2.26) | 245.15 (1.20) | 11.66 | 109.71 (1.31) | 77.57 (2.18) | **-29.30** |
| move-ego-180-2 | move-ego-low-0-0 | 219.56 (2.26) | 244.17 (7.74) | 11.21 | 12.15 (0.42) | 5.84 (0.42) | **-51.93** |
| move-ego-180-4 | crawl-0.4-0-d | 206.49 (0.97) | 206.49 (0.97) | 0.00 | 0.00 (0.00) | 0.00 (0.00) | NaN |
| move-ego-180-4 | move-ego-90-2 | 206.49 (0.97) | 184.61 (2.09) | -10.60 | 112.77 (1.15) | 65.99 (1.19) | **-41.48** |
| move-ego-180-4 | move-ego-low-0-0 | 206.49 (0.97) | 178.90 (9.01) | -13.36 | 8.71 (3.28) | 4.39 (1.61) | **-49.60** |
| move-ego-low-0-2 | crawl-0.4-0-d | 239.49 (14.67) | 239.49 (14.67) | 0.00 | 0.00 (0.00) | 0.00 (0.00) | NaN |
| move-ego-low-0-2 | move-ego-90-2 | 239.49 (14.67) | 189.77 (17.41) | -20.76 | 67.26 (6.48) | 41.67 (0.63) | **-38.05** |
| move-ego-low-0-2 | move-ego-low-0-0 | 239.49 (14.67) | 162.83 (35.96) | -32.01 | 30.70 (4.88) | 13.43 (3.89) | **-56.25** |

Table 6: Full evaluation results for all task pairs. Rewards are shown as mean (std) over 5 seeds. Percent change ($\Delta$) is relative to $z_1$.

| Task1 | Task2 | $z_1$ Env 1 | IIP Env 1 | %$\Delta$ Env 1 | $z_1$ Env 2 | IIP Env 2 | %$\Delta$ Env 2 |
|---|---|---|---|---|---|---|---|
| move-ego-low–90-2 | crawl-0.4-0-d | 241.15 (4.27) | 241.15 (4.27) | 0.00 | 0.00 (0.00) | 0.00 (0.00) | NaN |
| move-ego-low–90-2 | move-ego-90-2 | 241.15 (4.27) | 222.77 (6.49) | -7.62 | 3.16 (1.00) | 3.45 (0.41) | 9.18 |
| move-ego-low–90-2 | move-ego-low-0-0 | 241.15 (4.27) | 122.85 (9.30) | -49.06 | 21.07 (2.23) | 14.55 (8.97) | **-30.94** |
| move-ego-low-90-2 | crawl-0.4-0-d | 239.31 (2.49) | 239.31 (2.49) | 0.00 | 0.00 (0.00) | 0.00 (0.00) | NaN |
| move-ego-low-90-2 | move-ego-90-2 | 239.31 (2.49) | 141.87 (5.55) | -40.72 | 174.57 (2.93) | 87.19 (1.70) | **-50.05** |
| move-ego-low-90-2 | move-ego-low-0-0 | 239.31 (2.49) | 150.83 (37.45) | -36.97 | 24.66 (2.96) | 9.22 (4.61) | **-62.61** |
| move-ego-low-180-2 | crawl-0.4-0-d | 105.25 (5.20) | 105.25 (5.20) | 0.00 | 0.00 (0.00) | 0.00 (0.00) | NaN |
| move-ego-low-180-2 | move-ego-90-2 | 105.25 (5.20) | 155.85 (11.92) | 48.08 | 136.02 (2.26) | 67.80 (1.09) | **-50.15** |
| move-ego-low-180-2 | move-ego-low-0-0 | 105.25 (5.20) | 44.45 (19.06) | -57.77 | 11.40 (1.54) | 6.46 (1.18) | **-43.33** |
| jump-2 | crawl-0.4-0-d | 39.00 (0.77) | 39.00 (0.77) | 0.00 | 0.00 (0.00) | 0.00 (0.00) | NaN |
| jump-2 | move-ego-90-2 | 39.00 (0.77) | 37.91 (0.64) | -2.79 | 53.34 (14.04) | 26.83 (7.17) | **-49.70** |
| jump-2 | move-ego-low-0-0 | 39.00 (0.77) | 31.09 (1.33) | -20.28 | 53.06 (2.51) | 26.89 (2.16) | **-49.32** |
| rotate-x–5-0.8 | crawl-0.4-0-d | 22.24 (4.07) | 22.78 (2.52) | 2.43 | 17.33 (1.70) | 8.90 (0.77) | **-48.64** |
| rotate-x–5-0.8 | move-ego-90-2 | 22.24 (4.07) | 13.10 (3.20) | -41.10 | 5.77 (0.65) | 12.76 (0.89) | 121.14 |
| rotate-x–5-0.8 | move-ego-low-0-0 | 22.24 (4.07) | 18.63 (4.86) | -16.23 | 15.82 (3.95) | 7.84 (1.96) | **-50.44** |
| rotate-x-5-0.8 | crawl-0.4-0-d | 0.90 (0.88) | 2.12 (1.74) | 135.56 | 13.11 (13.44) | 6.84 (7.17) | **-47.83** |
| rotate-x-5-0.8 | move-ego-90-2 | 0.90 (0.88) | 0.90 (0.61) | 0.00 | 17.50 (12.82) | 14.40 (4.24) | **-17.71** |
| rotate-x-5-0.8 | move-ego-low-0-0 | 0.90 (0.88) | 1.20 (1.38) | 33.33 | 19.07 (15.53) | 10.75 (9.88) | **-43.63** |
| rotate-y–5-0.8 | crawl-0.4-0-d | 171.02 (16.23) | 171.02 (16.23) | 0.00 | 0.00 (0.00) | 0.00 (0.00) | NaN |
| rotate-y–5-0.8 | move-ego-90-2 | 171.02 (16.23) | 264.35 (3.40) | 54.57 | 25.95 (2.07) | 13.36 (1.24) | **-48.52** |
| rotate-y–5-0.8 | move-ego-low-0-0 | 171.02 (16.23) | 244.61 (6.41) | 43.03 | 69.36 (6.69) | 34.14 (3.58) | **-50.78** |
| rotate-y-5-0.8 | crawl-0.4-0-d | 145.96 (10.27) | 145.96 (10.27) | 0.00 | 0.00 (0.00) | 0.00 (0.00) | NaN |
| rotate-y-5-0.8 | move-ego-90-2 | 145.96 (10.27) | 252.02 (1.32) | 72.66 | 64.40 (2.53) | 32.18 (1.29) | **-50.03** |
| rotate-y-5-0.8 | move-ego-low-0-0 | 145.96 (10.27) | 258.72 (1.45) | 77.25 | 37.28 (1.83) | 27.64 (1.74) | **-25.86** |
| rotate-z–5-0.8 | crawl-0.4-0-d | 90.48 (13.77) | 91.11 (14.75) | 0.70 | 0.41 (0.28) | 0.22 (0.12) | **-46.34** |
| rotate-z–5-0.8 | move-ego-90-2 | 90.48 (13.77) | 84.88 (17.92) | -6.19 | 14.84 (1.92) | 15.85 (2.21) | 6.81 |
| rotate-z–5-0.8 | move-ego-low-0-0 | 90.48 (13.77) | 92.50 (16.40) | 2.23 | 2.93 (0.95) | 2.11 (0.93) | **-27.99** |
| rotate-z-5-0.8 | crawl-0.4-0-d | 134.40 (1.95) | 134.40 (1.95) | 0.00 | 0.15 (0.01) | 0.15 (0.01) | 0.00 |
| rotate-z-5-0.8 | move-ego-90-2 | 134.40 (1.95) | 129.26 (1.12) | -3.82 | 11.99 (0.39) | 10.80 (0.36) | **-9.92** |
| rotate-z-5-0.8 | move-ego-low-0-0 | 134.40 (1.95) | 141.96 (3.00) | 5.63 | 2.10 (0.95) | 2.40 (0.98) | 14.29 |
| raisearms-l-l | crawl-0.4-0-d | 272.33 (0.19) | 272.33 (0.19) | 0.00 | 0.00 (0.00) | 0.00 (0.00) | NaN |
| raisearms-l-l | move-ego-90-2 | 272.33 (0.19) | 273.08 (0.11) | 0.28 | 7.93 (0.47) | 50.73 (5.06) | 539.72 |
| raisearms-l-l | move-ego-low-0-0 | 272.33 (0.19) | 272.33 (0.19) | 0.00 | 0.00 (0.00) | 0.00 (0.00) | NaN |
| raisearms-l-m | crawl-0.4-0-d | 223.13 (37.29) | 223.13 (37.29) | 0.00 | 0.00 (0.00) | 0.00 (0.00) | NaN |
| raisearms-l-m | move-ego-90-2 | 223.13 (37.29) | 254.01 (7.09) | 13.84 | 7.31 (0.64) | 6.37 (3.66) | **-12.86** |
| raisearms-l-m | move-ego-low-0-0 | 223.13 (37.29) | 223.13 (37.29) | 0.00 | 0.00 (0.00) | 0.00 (0.00) | NaN |

Table 6: Full evaluation results for all task pairs. Rewards are shown as mean (std) over 5 seeds. Percent change ($\Delta$) is relative to $z_1$.

| Task1 | Task2 | $z_1$ Env 1 | IIP Env 1 | %$\Delta$ Env 1 | $z_1$ Env 2 | IIP Env 2 | %$\Delta$ Env 2 |
|---|---|---|---|---|---|---|---|
| raisearms-l-h | crawl-0.4-0-d | 230.75 (15.35) | 230.75 (15.35) | 0.00 | 0.00 (0.00) | 0.00 (0.00) | NaN |
| raisearms-l-h | move-ego-90-2 | 230.75 (15.35) | 254.26 (11.65) | 10.19 | 34.69 (22.53) | 18.11 (15.02) | **-47.79** |
| raisearms-l-h | move-ego-low-0-0 | 230.75 (15.35) | 230.75 (15.35) | 0.00 | 0.00 (0.00) | 0.00 (0.00) | NaN |
| raisearms-m-l | crawl-0.4-0-d | 100.93 (33.41) | 100.93 (33.41) | 0.00 | 0.00 (0.00) | 0.00 (0.00) | NaN |
| raisearms-m-l | move-ego-90-2 | 100.93 (33.41) | 80.59 (20.51) | -20.15 | 8.99 (1.65) | 4.46 (0.82) | **-50.39** |
| raisearms-m-l | move-ego-low-0-0 | 100.93 (33.41) | 100.93 (33.41) | 0.00 | 0.00 (0.00) | 0.00 (0.00) | NaN |
| raisearms-m-m | crawl-0.4-0-d | 258.61 (15.60) | 258.61 (15.60) | 0.00 | 0.00 (0.00) | 0.00 (0.00) | NaN |
| raisearms-m-m | move-ego-90-2 | 258.61 (15.60) | 262.61 (12.41) | 1.55 | 10.53 (6.77) | 5.27 (3.40) | **-49.95** |
| raisearms-m-m | move-ego-low-0-0 | 258.61 (15.60) | 258.61 (15.60) | 0.00 | 0.00 (0.00) | 0.00 (0.00) | NaN |
| raisearms-m-h | crawl-0.4-0-d | 180.80 (58.93) | 180.80 (58.93) | 0.00 | 0.00 (0.00) | 0.00 (0.00) | NaN |
| raisearms-m-h | move-ego-90-2 | 180.80 (58.93) | 243.77 (26.47) | 34.83 | 32.99 (16.97) | 16.53 (8.50) | **-49.89** |
| raisearms-m-h | move-ego-low-0-0 | 180.80 (58.93) | 180.80 (58.93) | 0.00 | 0.00 (0.00) | 0.00 (0.00) | NaN |
| raisearms-h-l | crawl-0.4-0-d | 46.61 (2.47) | 46.61 (2.47) | 0.00 | 0.00 (0.00) | 0.00 (0.00) | NaN |
| raisearms-h-l | move-ego-90-2 | 46.61 (2.47) | 47.46 (3.47) | 1.82 | 28.54 (7.75) | 16.45 (4.65) | **-42.36** |
| raisearms-h-l | move-ego-low-0-0 | 46.61 (2.47) | 46.61 (2.47) | 0.00 | 0.00 (0.00) | 0.00 (0.00) | NaN |
| raisearms-h-m | crawl-0.4-0-d | 132.29 (67.74) | 132.29 (67.74) | 0.00 | 0.00 (0.00) | 0.00 (0.00) | NaN |
| raisearms-h-m | move-ego-90-2 | 132.29 (67.74) | 152.65 (35.98) | 15.39 | 23.48 (20.17) | 11.73 (10.07) | **-50.04** |
| raisearms-h-m | move-ego-low-0-0 | 132.29 (67.74) | 132.29 (67.74) | 0.00 | 0.00 (0.00) | 0.00 (0.00) | NaN |
| raisearms-h-h | crawl-0.4-0-d | 229.12 (31.25) | 229.12 (31.25) | 0.00 | 0.00 (0.00) | 0.00 (0.00) | NaN |
| raisearms-h-h | move-ego-90-2 | 229.12 (31.25) | 256.48 (6.88) | 11.94 | 13.58 (2.59) | 7.40 (1.72) | **-45.51** |
| raisearms-h-h | move-ego-low-0-0 | 229.12 (31.25) | 229.12 (31.25) | 0.00 | 0.00 (0.00) | 0.00 (0.00) | NaN |
| crouch-0 | crawl-0.4-0-d | 243.32 (4.23) | 243.32 (4.23) | 0.00 | 0.00 (0.00) | 0.00 (0.00) | NaN |
| crouch-0 | move-ego-90-2 | 243.32 (4.23) | 248.32 (1.71) | 2.05 | 19.94 (1.20) | 12.59 (2.33) | **-36.86** |
| crouch-0 | move-ego-low-0-0 | 243.32 (4.23) | 236.62 (5.14) | -2.75 | 102.88 (0.27) | 101.87 (1.46) | **-0.98** |
| sitonground | crawl-0.4-0-d | 201.78 (13.35) | 201.78 (13.35) | 0.00 | 0.00 (0.00) | 0.00 (0.00) | NaN |
| sitonground | move-ego-90-2 | 201.78 (13.35) | 217.28 (1.59) | 7.68 | 7.73 (2.71) | 8.88 (2.81) | 14.88 |
| sitonground | move-ego-low-0-0 | 201.78 (13.35) | 52.48 (44.14) | -73.99 | 112.13 (7.98) | 26.75 (19.17) | **-76.14** |
| lieonground-up | crawl-0.4-0-d | 216.19 (1.07) | 216.19 (1.07) | 0.00 | 0.00 (0.00) | 0.00 (0.00) | NaN |
| lieonground-up | move-ego-90-2 | 216.19 (1.07) | 177.86 (7.23) | -17.73 | 3.82 (0.91) | 7.86 (0.60) | 105.76 |
| lieonground-up | move-ego-low-0-0 | 216.19 (1.07) | 0.00 (0.00) | -100.00 | 61.58 (0.66) | 30.78 (0.31) | **-50.02** |
| lieonground-down | crawl-0.4-0-d | 203.05 (2.94) | 103.78 (30.73) | -48.89 | 142.72 (0.22) | 64.29 (19.94) | **-54.95** |
| lieonground-down | move-ego-90-2 | 203.05 (2.94) | 215.02 (0.94) | 5.90 | 2.88 (0.17) | 4.98 (0.56) | 72.92 |
| lieonground-down | move-ego-low-0-0 | 203.05 (2.94) | 194.56 (4.75) | -4.18 | 52.82 (0.79) | 38.18 (1.56) | **-27.72** |
| split-0.5 | crawl-0.4-0-d | 251.20 (3.19) | 251.20 (3.19) | 0.00 | 0.00 (0.00) | 0.00 (0.00) | NaN |
| split-0.5 | move-ego-90-2 | 251.20 (3.19) | 234.50 (7.18) | -6.65 | 6.90 (1.57) | 13.44 (0.82) | 94.78 |
| split-0.5 | move-ego-low-0-0 | 251.20 (3.19) | 180.68 (20.64) | -28.07 | 108.87 (0.22) | 94.77 (6.85) | **-12.95** |

Table 6: Full evaluation results for all task pairs. Rewards are shown as mean (std) over 5 seeds. Percent change ($\Delta$) is relative to $z_1$.

| Task1 | Task2 | $z_1$ Env 1 | IIP Env 1 | %$\Delta$ Env 1 | $z_1$ Env 2 | IIP Env 2 | %$\Delta$ Env 2 |
|---|---|---|---|---|---|---|---|
| split-1 | crawl-0.4-0-d | 94.13 (27.83) | 94.13 (27.83) | 0.00 | 0.00 (0.00) | 0.00 (0.00) | NaN |
| split-1 | move-ego-90-2 | 94.13 (27.83) | 56.11 (52.51) | -40.39 | 26.40 (16.74) | 16.93 (5.42) | **-35.87** |
| split-1 | move-ego-low-0-0 | 94.13 (27.83) | 12.65 (15.86) | -86.56 | 107.55 (24.42) | 63.64 (26.50) | **-40.83** |
| crawl-0.4-0-u | crawl-0.4-0-d | 108.70 (4.52) | 108.70 (4.52) | 0.00 | 0.00 (0.00) | 0.00 (0.00) | NaN |
| crawl-0.4-0-u | move-ego-90-2 | 108.70 (4.52) | 93.90 (5.57) | -13.62 | 6.15 (0.16) | 3.34 (0.30) | **-45.69** |
| crawl-0.4-0-u | move-ego-low-0-0 | 108.70 (4.52) | 87.81 (4.90) | -19.22 | 65.32 (0.64) | 91.75 (1.63) | 40.46 |
| crawl-0.4-2-u | crawl-0.4-0-d | 15.66 (5.75) | 15.66 (5.75) | 0.00 | 0.00 (0.00) | 0.00 (0.00) | NaN |
| crawl-0.4-2-u | move-ego-90-2 | 15.66 (5.75) | 13.10 (2.39) | -16.35 | 5.97 (1.53) | 4.50 (1.68) | **-24.62** |
| crawl-0.4-2-u | move-ego-low-0-0 | 15.66 (5.75) | 17.77 (2.20) | 13.47 | 70.55 (3.62) | 88.67 (3.51) | 25.68 |
| crawl-0.5-0-u | crawl-0.4-0-d | 108.26 (5.00) | 108.26 (5.00) | 0.00 | 0.00 (0.00) | 0.00 (0.00) | NaN |
| crawl-0.5-0-u | move-ego-90-2 | 108.26 (5.00) | 93.68 (5.36) | -13.47 | 6.12 (0.17) | 3.10 (0.09) | **-49.35** |
| crawl-0.5-0-u | move-ego-low-0-0 | 108.26 (5.00) | 87.82 (6.00) | -18.88 | 65.30 (0.81) | 92.37 (1.81) | 41.45 |
| crawl-0.5-2-u | crawl-0.4-0-d | 16.02 (5.50) | 16.02 (5.50) | 0.00 | 0.00 (0.00) | 0.00 (0.00) | NaN |
| crawl-0.5-2-u | move-ego-90-2 | 16.02 (5.50) | 12.25 (2.32) | -23.53 | 6.09 (1.44) | 5.12 (1.91) | **-15.93** |
| crawl-0.5-2-u | move-ego-low-0-0 | 16.02 (5.50) | 18.49 (3.02) | 15.42 | 70.64 (3.77) | 91.72 (6.83) | 29.84 |
| crawl-0.4-0-d | crawl-0.5-2-u | 218.88 (12.18) | 218.88 (12.18) | 0.00 | 0.00 (0.00) | 0.00 (0.00) | NaN |
| crawl-0.4-0-d | move-ego-90-2 | 218.88 (12.18) | 146.25 (29.70) | -33.18 | 6.73 (1.49) | 5.40 (1.19) | **-19.76** |
| crawl-0.4-0-d | move-ego-low-0-0 | 218.88 (12.18) | 169.85 (18.72) | -22.40 | 142.74 (8.28) | 72.01 (6.09) | **-49.55** |
| crawl-0.4-2-d | crawl-0.5-2-u | 10.81 (4.87) | 10.81 (4.87) | 0.00 | 0.00 (0.00) | 0.00 (0.00) | NaN |
| crawl-0.4-2-d | move-ego-90-2 | 10.81 (4.87) | 17.04 (7.20) | 57.63 | 5.97 (1.43) | 4.43 (1.44) | **-25.80** |
| crawl-0.4-2-d | move-ego-low-0-0 | 10.81 (4.87) | 19.02 (4.67) | 75.95 | 126.65 (45.26) | 65.27 (18.68) | **-48.46** |
| crawl-0.5-0-d | crawl-0.5-2-u | 156.95 (28.19) | 156.95 (28.19) | 0.00 | 0.00 (0.00) | 0.00 (0.00) | NaN |
| crawl-0.5-0-d | move-ego-90-2 | 156.95 (28.19) | 152.07 (42.10) | -3.11 | 5.88 (0.53) | 5.39 (2.35) | **-8.33** |
| crawl-0.5-0-d | move-ego-low-0-0 | 156.95 (28.19) | 123.45 (15.56) | -21.34 | 86.71 (14.81) | 44.84 (8.85) | **-48.29** |
| crawl-0.5-2-d | crawl-0.5-2-u | 25.41 (11.08) | 25.41 (11.08) | 0.00 | 0.00 (0.00) | 0.00 (0.00) | NaN |
| crawl-0.5-2-d | move-ego-90-2 | 25.41 (11.08) | 26.20 (20.26) | 3.11 | 5.99 (3.73) | 6.45 (1.81) | 7.68 |
| crawl-0.5-2-d | move-ego-low-0-0 | 25.41 (11.08) | 25.13 (9.74) | -1.10 | 83.29 (30.29) | 45.65 (14.72) | **-45.19** |

## A.5 BODY SEGMENTS AND KINEMATIC VARIABLES

The environment's reward signal is composed of a set of basic elements whose values are determined by body-part descriptors and their temporal changes. The body is partitioned into the *trunk* and the *limbs*, and four types of kinematic variables describe state changes.

**Trunk** `Pelvis`, `Torso`, `Spine`, `Chest`, `Neck`, `Head`.

**Limbs** `L_Hip`, `R_Hip` (left/right hip), `L_Knee`, `R_Knee` (left/right knee), `L_Ankle`, `R_Ankle` (left/right ankle; cf. `rewards.py:548--549`), `L_Toe`, `R_Toe` (left/right toe), `L_Thorax`, `R_Thorax` (left/right thorax), `L_Hand`, `R_Hand` (left/right hand), `L_Shoulder`, `R_Shoulder` (left/right shoulder), `L_Elbow`, `R_Elbow` (left/right elbow), `L_Wrist`, `R_Wrist` (left/right wrist).

**Kinematic variables** `pos` (position), `rot` (orientation/rotation), `vel` (linear velocity), `ang` (angular velocity).

Accordingly, all reward terms are constructed from the above body-part features and kinematic variables. By specifying appropriate subsets of parts and change variables when defining the reward, the agent is incentivized to realize the desired behaviors.

## A.6 FORMULA DESCRIPTION

**Behavior definitions.** Each behavior $S$ is composed of multiple action primitives. The following formulas present the symbolic specification of each behavior; the corresponding textual descriptions of the action primitives are summarized in Table 7.

### LOCOMOTIONREWARD

Encourages a humanoid to move at a prescribed speed and heading. The reward comprises standing height, torso uprightness, translational speed, and heading control, and supports egocentric targets as well as low-posture locomotion.

$$S_{\text{LocomotionReward}} = \begin{cases} s_{\text{small\_control}} \cdot s_{\text{stand\_reward}} \cdot s_{\text{dont\_move}}, & \text{if } move\_speed = 0 \\ s_{\text{small\_control}} \cdot s_{\text{stand\_reward}} \cdot s_{\text{move}} \cdot s_{\text{angle\_reward}}, & \text{if } move\_speed \neq 0 \end{cases}$$

### JUMPREWARD

Encourages the agent to jump to a specified height. Performance is assessed by combining head height, torso uprightness, and upward velocity.

$$S_{\text{JumpReward}} = s_{\text{jumping}} \cdot s_{\text{upright}} \cdot s_{\text{up\_velocity}}$$

### HEADSTANDREWARD

Encourages execution of a headstand. Evaluation considers pelvis elevation, global body orientation, foot placement, and verified head–ground contact.

$$S_{\text{HeadstandReward}} = s_{\text{height\_reward}} \cdot s_{\text{small\_control}} \cdot s_{\text{headstand}} \cdot s_{\text{dont\_move}} \cdot s_{\text{dont\_rotate}} \cdot s_{\text{high\_left\_foot}} \cdot s_{\text{high\_right\_foot}} \cdot s_{\text{high\_head}}$$

### ROTATIONREWARD

Encourages rotation about a specified axis at a target angular velocity. The objective aggregates height maintenance, rotational speed, and whole-body alignment.

$$S_{\text{RotationReward}} = s_{\text{move}} \cdot s_{\text{height\_reward}} \cdot s_{\text{small\_control}} \cdot s_{\text{aligned}}$$

### ARMSREWARD

Encourages raising the arms to designated heights. The criterion checks whether the left and right hands reach predefined low, medium, and high bands.

$$S_{\text{ArmsReward}} = s_{\text{small\_control}} \cdot s_{\text{stand\_reward}} \cdot s_{\text{dont\_move}} \cdot s_{\text{left\_arm}} \cdot s_{\text{right\_arm}}$$

### LIEDOWNREWARD

Encourages lying on the ground. Assessment inspects ground contact of key segments, chest orientation, and overall body alignment.

$$S_{\text{LieDownReward}} = s_{\text{small\_control}} \cdot s_{\text{ground\_reward}} \cdot s_{\text{dont\_move}} \cdot s_{\text{orient\_reward}}$$

### SPLITREWARD

Encourages a split posture. Evaluation considers ankle separation distance, pelvic position, and head height.

$$S_{\text{SplitReward}} = s_{\text{head\_rew}} \cdot s_{\text{split\_rew}} \cdot s_{\text{pelvis\_pos}} \cdot s_{\text{dont\_move}} \cdot s_{\text{small\_control}}$$

Table 7: Descriptions of sub-rewards $s$ used in reward function definitions.

| Sub-reward | Description |
|---|---|
| $s_{small\_control}$ | Penalizes large control inputs, encouraging smooth and small torques. Helps reduce jitter and unnecessary energy consumption. |
| $s_{stand\_reward}$ | Ensures proper head height and torso uprightness, encouraging the agent to maintain a standing posture. |
| $s_{dont\_move}$ | Rewards staying still when no movement is required by limiting horizontal center-of-mass velocity. Prevents unnecessary swaying. |
| $s_{move}$ | Encourages the center-of-mass velocity to match the target speed, and aligns with a specified direction if given. Ensures proper forward motion. |
| $s_{angle\_reward}$ | Measures alignment between current velocity direction and the target direction. High reward when aligned, lower when deviating. |
| $s_{jumping}$ | Rewards if the head height exceeds a desired jump threshold, ensuring the jumping task is achieved. |
| $s_{upright}$ | Encourages torso uprightness, preventing excessive bending or collapsing. Determined mainly by chest orientation. |
| $s_{up\_velocity}$ | Encourages upward velocity of the center of mass or head, supporting jumping and lifting motions. |
| $s_{height\_reward}$ | Ensures pelvis or body parts maintain a proper height range, discouraging collapse or being too low. |
| $s_{headstand}$ | Encourages pelvis inversion aligned with a head-supported pose, forming a headstand. Defined mainly by pelvis orientation. |
| $s_{dont\_rotate}$ | Penalizes large angular velocities, encouraging stability and reducing erratic spinning. |
| $s_{high\_left\_foot}$, $s_{high\_right\_foot}$ | Ensures the feet are lifted above the ground during specific poses (e.g., headstand). Prevents contact with the ground. |
| $s_{high\_head}$ | Keeps the head safely above ground level, avoiding collapse or head contact with the floor. |
| $s_{aligned}$ | Encourages pelvis orientation to match the target axis. Important in rotation tasks. |
| $s_{left\_arm}$, $s_{right\_arm}$ | Ensures the arms are raised to specified height ranges (low, medium, high, extended). |
| $s_{ground\_reward}$ | Encourages body parts to stay close to the ground, used in lying or prone tasks. |
| $s_{orient\_reward}$ | Ensures body geometry is oriented consistently with the target direction (e.g., torso or limbs aligned). |
| $s_{split\_rew}$ | Encourages the distance between legs to exceed a threshold, used in split or straddle tasks. |
| $s_{pelvis\_pos}$ | Ensures pelvis position stays near the ground or within a proper height range. |
| $s_{head\_rew}$ | Encourages the head to stay at a reasonable height, avoiding collapse. |
| $s_{pelvis\_reward}$ | Keeps pelvis height within a desired range, important for sitting or crouching tasks. |
| $s_{knee\_reward}$ | Based on whether knees touch or avoid the ground. Controls knee support conditions. |
| $s_{alignment\_reward}$ | Measures alignment of multiple body parts' orientations, encouraging coordinated posture. |
| $s_{arms}$ | Sub-reward derived from arm-related tasks (e.g., raising arms). Part of ArmsReward. |
| $s_{locomotion}$ | Sub-reward derived from locomotion-related tasks (e.g., speed, direction). Part of LocomotionReward. |

SITONGROUNDREWARD

Encourages sitting on the ground or adopting a squat posture. The metric evaluates pelvic height, head position, knee configuration, and torso uprightness.

$$S_{SitOnGroundReward} = s_{small\_control} \cdot s_{stand\_reward} \cdot s_{dont\_move} \cdot s_{pelvis\_reward} \cdot s_{knee\_reward}$$

While estimating the scalar parameter $\lambda$ via the bisection method, we found that injecting the representation $z_1$ produced by the *Locomotion* task into the *SitOnGround* environment yielded a reward of approximately $0.13$. This behavior stems from seated–height constraints imposed by `stand_reward` and `pelvis_reward`, which suppress the signal and induce premature convergence of the bisection procedure. To address this, we ablated these two terms and defined a revised objective, *SitOnGround v2*, which restored an informative optimization landscape; the bisection iterations then proceeded to convergence, yielding $\lambda = 1$.

$$S_{\text{SitOnGroundReward-v2}} = s_{\text{small\_control}} \cdot s_{\text{dont\_move}} \cdot s_{\text{knee\_reward}}$$

CRAWLREWARD

Encourages crawling at a specified body height and speed. The objective considers spinal height, body orientation, translational speed, and angular alignment.

$$S_{\text{CrawlReward}} = \begin{cases} s_{\text{dont\_move}} \cdot s_{\text{alignment\_reward}}, & \text{if } move\_speed = 0 \\ s_{\text{alignment\_reward}} \cdot s_{\text{move}} \cdot s_{\text{angle\_reward}}, & \text{if } move\_speed \neq 0 \end{cases}$$

MOVEANDRAISEARMSREWARD

A composite task coupling locomotion with arm-raising. It internally combines *LocomotionReward* and *ArmsReward* and modulates the locomotion coefficient based on arm posture.

$$S_{\text{MoveAndRaiseArmsReward}} = \frac{\alpha \, s_{\text{arms}} + \beta \, s_{\text{locomotion}}}{\alpha + \beta}, \quad \text{where } \alpha = \texttt{arm\_coeff}, \ \beta = \texttt{loc\_coeff}$$

