# OpenReview forum: "Controlling Zero-Shot Agents Via Inverse Inverse Planning"
_ICLR.cc/2026/Conference — ICLR 2026 Conference Withdrawn Submission_

### Official Review · Reviewer_efjq · 2025-10-25

**Soundness:** 2
**Presentation:** 1
**Contribution:** 2
**Rating:** 2
**Confidence:** 4

**Summary:**

This paper presents an approach to removing undesired behavioral traits from pre-trained policies. The method combines the forward–backward representation approach, which leverages task embeddings to enable zero-shot generalization to unseen tasks, with inverse inverse planning, which produces behaviors that are easier to interpret.

**Strengths:**

The general idea of improving the interpretability of zero-shot generalization agents using inverse inverse planning seems interesting and promising, and it is a direction worth pursuing.

The related work, key methods, and background material are extensively discussed and provide sufficient context to understand the proposed approach.

**Weaknesses:**

## Applicability
The original idea of IIP is to obtain plans or policies that maximize the probability that an observer infers the correct goal from a perceived state–action trajectory. The approach outlined in this paper attempts to achieve this indirectly by removing certain behavioral traits from an existing policy. The drawback of this approach is that one must (1) know which behavioral feature to remove and (2) already have access to a policy or at least a pre-collected dataset thereof. This can be a significant limitation in the case of more complex behaviors or when additivity does not hold. One such instance is illustrated in Figure 5: there is no immediate or intuitive connection between the removal of a sitting behavior and the emergence of a more natural walking gait.

## Clarity and Description of the Approach
The paper devotes a substantial portion of its space to introducing prior methods, which is not necessarily a bad choice. Unfortunately, the main section, where the proposed approach should be described, lacks a clear and detailed explanation of the algorithm. This leaves out many important details, such as a more thorough description of the individual components. For instance, it would be interesting to know why the forward–backward approach was chosen over other types of task embeddings, or how critical the bisection procedure is for obtaining a suitable $\lambda$.

## Experimental Evaluation
Including ablation studies for the individual components of the algorithm would strengthen the paper by highlighting the importance of specific design decisions, such as using the forward–backward approach or the bisection method.

**Section 5.1**

The main focus of the experimental evaluation is to demonstrate that applying the proposed method reduces the reward in one task while preserving the reward in another. While this can indeed illustrate the removal of certain behaviors in most tasks, the paper lacks a discussion of task configurations where this approach might fail.

The plots in section 5.1 show large variance in the rewards. When claiming that “IIP generally preserves performance,” it would be more convincing to include an aggregate measure, such as the $R^2$ score, or an analysis of the relationship between performance loss in task A versus task B. This would add more depth to the results.
Furthermore, for Tables 1–4: instead of showing both the top 5 percentages and the top 5 absolute values, a single table would suffice. The freed space could be used for an additional experiment.

**Sections 5.2–5.4**

The results presented in these sections are mostly anecdotal. While such examples can be useful to illustrate subjective features of certain behaviors, they take up too much emphasis in the current form. The paper would benefit from shortening this part or moving it to the appendix, allowing for a more detailed description of the algorithm or additional ablation studies.

**Questions:**

- Why did you choose the forward-backward representation over other methods?
- The tables show that significant drops in performance in the second task come hand-in-hand with drastic performance drops in the other task. How does this influence the original, visually perceived, behavior?

---

### Official Review · Reviewer_DNJ8 · 2025-10-27

**Soundness:** 3
**Presentation:** 4
**Contribution:** 3
**Rating:** 4
**Confidence:** 4

**Summary:**

This paper presents a method for improving zero-shot RL agents by increasing the interpretability of the policy through fine-tuning the latent task vector. The paper introduces the notion of inverse-inverse planning that builds on the inverse planning framework. In their formulation, the agent attempts to increase the observer's posterior probability of inferring its goal by adjusting its policy. Rather than search in the action-space, this method does a search in the latent embedding space of the ZSRL agent. The candidate solution upon which they perform a bisection search, to their optimization problem is a sum of two latent embeddings, one of which describes the desired behavior and the other which describes the undesired behavior. The paper demonstrates a series of quantitative and qualitative successes and failure cases of the method on the metamotivo benchmark.

**Strengths:**

This paper is written very clearly and is well motivated. It presents a simple algorithm for searching for latent task embeddings for a desired task. The method leverages pre-trained behavior foundation models that have a large capacity for procuding diverse behavior. The paper presents key results on the successes of the method and importantly acknowledges where the method fails.

The bisection search provides a simple solution to a computationally hard problem (inverse planning).

**Weaknesses:**

There are several weakness that must be addressed.

1. **Structure of** $z_\text{iip}$: In many implementations of the forward-backward representation, the task embedding is restricted to the $l_2$ norm ball. However, the construction of $z_\text{iip} = z_1 - \lambda z_2$ violates this norm. Is this consistent with the metamotivo pre-trained models?

2. **Key baselines missing**: It is not clear that the method was compared against any known baseline. For instance, if the reward functions (not solely state distributions) are known, then the most naive baseline would be to compare to a vanilla RL agent trained from scratch on the tasks. Further, there are other works that present methods for doing search in the latent space of BFMs such as Fast Adaptation with Behavioral Foundation Models [Sikchi 2025]. These methods could be easily adapted to act as common sense baselines to this work. Is there another reason that these baselines are not present in the work?

3. **How are the desired and undesired task embeddings initially found?**: It is not clear exactly how the task embeddings $z1$ and $z_2$ are calculated in the first place. If $z_1$ is not fully descriptive of the true reward function for task 1, can you not sample the states used to calculate $z_1$ more densely? It's clear from the failure cases that when there is significant behavioral overlap between task 1 and task 2, it is insufficient to do just the bisection search and another method is necessary. I am mostly curious how the initial task embeddings are have so much undesired behavior to begin with.

4. **Other zero-shot RL methods**: Does this method work on other zero-shot RL methods such as PSM [Agarwal 2025] or HILP [Park 2024]


[Sikchi, et al. 2025] Fast Adaptation with Behavioral Foundation Models, Harshit Sikchi and Andrea Tirinzoni and Ahmed Touati and Yingchen Xu and Anssi Kanervisto and Scott Niekum and Amy Zhang and Alessandro Lazaric and Matteo Pirotta

[Agarwal et al. 2025] Proto Successor Measure: Representing the Behavior Space of an RL Agent, Siddhant Agarwal and Harshit Sikchi and Peter Stone and Amy Zhang

[Park et al. 2024] Foundation Policies with Hilbert Representations, Seohong Park, Tobias Kreiman, Sergey Levine

**Questions:**

Along with the questions I ask in the weaknesses section, please answer the following:

1. Is it possible to do this bisection search iteratively for several undesired tasks?
2. Does this method apply to other behavior foundation policy methods?
3. Can the authors include comparisons to basic common sense alternatives to fine-tuning task embeddings in this setting?

---

### Official Review · Reviewer_LmQu · 2025-10-28

**Soundness:** 1
**Presentation:** 2
**Contribution:** 1
**Rating:** 0
**Confidence:** 4

**Summary:**

The paper proposes to integrate zero-shot reinforcement learning with inverse inverse planning. Specifically, for a given task $z$, the idea is to remove behaviors that are specific to some other task $z'$ without affecting the agent's performance on the original task $z$. The intention is to improve the "readability" of the agent's behaviors so that another agent (such as a human) can better understand that the agent is attempting to pursue task $z$.

**Strengths:**

1. The introduction is clear and flows well.
2. The paper provides a good coverage of related work.
3. The IIP bisection algorithm is very clearly described.

**Weaknesses:**

The following weaknesses are roughly ordered in terms of importance.

1. The primary contribution of the paper -- the "IIP via bisection" algorithm -- is trivial (and potentially not well-founded; see questions in the next box). The proposed idea is to conduct bisection search over the parameter $\lambda$ to find a setting that sufficiently decreases the agent's performance (reward) on a specified secondary (undesired) task $z_2$ by forcing the agent to act on a modified task $z_{iip} = z_1 - \lambda z_2$.
    - This could easily be phrased instead as an optimization process (to find some $z'$ that maximizes reward on task $z_1$ while minimizing reward on task $z_2$), which would be more general and perhaps yield better results. Of course, this would not be as efficient as bisection search, but the appropriate use of bisection search requires several additional assumptions which do not seem to hold in the context of this paper.
    - Even though the proposed method is simple, it is possible that a paper could be made that contains nontrivial results showing the importance of the method and shedding light on these topics (ZSRL, IIP). But, that is not this paper; the quantitative results are weak (25%-35% failure rate, depending on definition of failure, or perhaps even higher if these two numbers are not fully correlated) and there is little empirical study of the interpretability aspect (see next point).
2. Although it is mentioned several times throughout the paper, including as claimed contributions in the introduction and conclusion, there is no clear evaluation and discussion of action/plan interpretability beyond (a) the use of the secondary task reward proxy and (b) the inclusion of videos for manual inspection. Since it is mentioned so often, and especially since the introduction specifically claims "the probability of inferring the intended goal increases", an extended discussion of this topic seems warranted.
    - This extends to the claims made in the conclusion about "resolving behavioral ambiguity" and "the ability to improve narrative coherence and physical plausibility in agent behaviors".
3. The introduction mentions "removing boundary-violating behaviors" as a contribution, but I do not recall seeing this mentioned anywhere else in the paper.
4. Line 309 mentions that the Metamotivo pretrained agents have "high-fidelity and naturalistic movement", which seems to imply that the current paper's goal of "adjusting animations to appear biomechanically natural" is redundant.
5. Lines 376-377 conjecture that "[...] some task embeddings are highly entangled", which makes IIP fail on certain task pairs, but this very interesting point is not explored or investigated any further.
6. Section 5.3 anthropomorphizes the agent's actions (related to the narrative aspect of IIP), but it seems that this is post-hoc: although the modified policies display behavior that can be interpreted in certain ways, the method in the paper does not provide any way to intentionally modify the agent's behavior in a way that leads to a specific interpretation (e.g., the given example of "to portray the agent as being afraid of bugs on the ground"). One could further argue that this detracts from the paper's claim of implementing IIP.
7. The claims at the end of 5.1 (lines 399-402) are not supported by the quantitative results, and the last claim about "[...] increases in intended-goal inference and reductions in unintended-goal attribution" were not even evaluated in any rigorous way. The paper uses a drop in secondary-task reward as a proxy for these factors, which is not enough to make such claims.
8. The notation used for FB formulas is not explained well; for example, the variable $z$ is not clearly explained as representing a task until section 4.
9. Certain aspects of the related work and preliminary exposition may be unnecessary for the overall story of the paper:
    - Equations 1, 2, and 3 do not seem important to any other part of the paper.
    - The discussion of IIP in 2.2 is interesting, but perhaps longer than necessary. Such a detailed historical summary of IP/IIP may be better suited to a longer-form article (such as for a journal).
10. Section 3.3 seems like it belongs somewhere in section 2, e.g., in section 2.2.
11. There are some rather strange statements that are made without explanation, e.g., line 297-298: "preventing sitting during walking to suggest aversion to bugs and evoke emotional resonance [...]". While the bug thing is explained later, at this point in the paper, it makes no sense.
12. The subfigures in Figure 4 are not labeled with the intended tasks (what are goal 1 and goal 2?).
13. Section 5.4 discusses "physical plausibility of agent motion". As noted in point 4, the pretrained agents supposedly already have "high-fidelity and naturalistic movement", so pursuing this seems redundant. In addition, there are other prior methods for improving the naturalness and stability of learned movement policies (such as energy/torque penalties), but they are not mentioned or discussed here.

**Questions:**

1. Since bisection search is used, is the reward signal guaranteed to be monotonic in $\lambda$? If so, is this proven somewhere (e.g., in prior work)? If not, what justifies the use of bisection search as opposed to a generic optimization method?
    - Additionally, why use bisection, which constrains the search to linearly interpolate between $z_1$ and $z_2$, instead of a more general search throughout the space of task embeddings? (obviously, bisection is more efficient, but it does not seem that there is any reason to suspect that a good solution must lie exactly on a line between $z_1$ and $z_2$)
2. The bisection algorithm has a hard cutoff on the number of iterations; why not use a threshold on $|\lambda_{max}-\lambda_{min}|$?

---

### Note · Authors · 2025-11-12

I have read and agree with the venue's withdrawal policy on behalf of myself and my co-authors.